# Crystallized Intelligence, Fluid Intelligence, and Need for Cognition: Their Longitudinal Relations in Adolescence

**DOI:** 10.3390/jintelligence12110104

**Published:** 2024-10-24

**Authors:** Vsevolod Scherrer, Moritz Breit, Franzis Preckel

**Affiliations:** Department of Psychology, Trier University, D-54286 Trier, Germany; breitm@uni-trier.de (M.B.); preckel@uni-trier.de (F.P.)

**Keywords:** crystallized intelligence, fluid intelligence, investment traits, need for cognition, investment theory

## Abstract

Investment theory and related theoretical approaches suggest a dynamic interplay between crystallized intelligence, fluid intelligence, and investment traits like need for cognition. Although cross-sectional studies have found positive correlations between these constructs, longitudinal research testing all of their relations over time is scarce. In our pre-registered longitudinal study, we examined whether initial levels of crystallized intelligence, fluid intelligence, and need for cognition predicted changes in each other. We analyzed data from 341 German students in grades 7–9 who were assessed twice, one year apart. Using multi-process latent change score models, we found that changes in fluid intelligence were positively predicted by prior need for cognition, and changes in need for cognition were positively predicted by prior fluid intelligence. Changes in crystallized intelligence were not significantly predicted by prior Gf, prior NFC, or their interaction, contrary to theoretical assumptions. This pattern of results was largely replicated in a model including all constructs simultaneously. Our findings support the notion that intelligence and investment traits, particularly need for cognition, positively interact during cognitive development, but this interplay was unexpectedly limited to Gf.

## 1. Bullet Points

Initial levels of fluid intelligence positively predicted changes in need for cognition, supporting the environmental success hypothesis.Initial levels of need for cognition positively predicted changes in fluid intelligence, supporting the environmental enrichment hypothesis.Initial levels of fluid intelligence or need for cognition did not significantly predict changes in crystallized intelligence.

## 2. Introduction

Intelligence is a central construct in psychology and related fields such as education, sociology, neuroscience, and economics ([37]; [43]). It is linked to key life outcomes such as health and longevity ([24]), academic performance ([53]; [73]), career success ([7]; [42]), socioeconomic achievements ([60]), and overall subjective well-being ([58]). Intelligence is often conceptualized as “the most stable psychological trait” ([47]). While previous longitudinal research has shown high rank-order stability over time for various facets of intelligence, its stability is lower during childhood and early adolescence ([10]). Consequently, individuals in these age groups show different developmental trends in intelligence, resulting in changing rank-order when comparing intelligence across different time points. To promote positive developmental trends in children and adolescents and to inform potential interventions that foster cognitive growth, it is crucial to understand the factors that influence individual change over time and the conditions that contribute to a positive development. A substantial body of research has been dedicated to uncovering such influencing factors, including extensive research on genetics (e.g., [21]; [64]) and environmental influences such as schooling (e.g., [11]; [51]; [65]). 

In this vein, another significant field of research investigates the developmental interplay between intelligence and personality. Several theories have been proposed regarding the relationships between crystallized intelligence, fluid intelligence, and so-called investment traits, which are assumed to motivate the investment of fluid intelligence into the acquisition of crystallized intelligence ([16]; [68]; [76]). Supporting the relevance of investment traits for performance domains, previous research has shown their significant impact on academic and career success (e.g., [39]; [62]; [36]). However, the relationship between investment traits and intelligence may not be unidirectional. Given that personality traits undergo substantial changes across development, demonstrated by their modest rank-order stability ([8]), life experiences impacted by one’s intelligence, such as academic successes or failures, may significantly influence their development. This highlights the importance of treating intelligence and investment traits as both outcomes and predictors in their developmental interplay (e.g., [76]).

A large body of cross-sectional research supports the hypothesis that interindividual differences in fluid intelligence, crystallized intelligence, and investment traits are interrelated (for a meta-analysis, see [67]). However, longitudinal research testing their interplay is scarce and inconclusive. To address this research gap, we conducted a longitudinal study with young adolescents who were assessed twice with a time interval of approximately one year. The assessments included measures of crystallized intelligence, fluid intelligence, and the investment trait of need for cognition.

### 2.1. Crystallized and Fluid Intelligence

The distinction between crystallized and fluid components of human intelligence originated with the influential work of [16] ([16]), and the definitions of these constructs have remained largely unchanged over time. Crystallized intelligence (Gc) refers to the ability to understand and utilize culturally valued knowledge, encompassing information and skills acquired through experience, education, and acculturation ([54]). According to Cattell, in adult life, peaks of performance are primarily determined by Gc, which can continue to increase well into adulthood and only decline slowly towards the end of life ([66]; [70]). Fluid intelligence (Gf) is defined as the application of purposeful and deliberate procedures to solve novel problems that cannot be resolved using previously acquired knowledge ([54]). According to [16] ([16]), Gf increases until young adulthood, reaches a plateau, and then declines over time ([66]). He suggested that Gf predominantly defines peak performance during childhood and adolescence.

Modern theories of the structure of intelligence recognize Gc and Gf as important facets of intelligence. According to the Cattell-Horn-Carroll model (CHC model; [40]; [54]), Gf and Gc are broad cognitive abilities positioned at Stratum 2, hierarchically below a g-factor of general intelligence at Stratum 1. Most intelligence tests include index scores for Gc and Gf (e.g., [55]; [71]) or even exclusively target these abilities (e.g., [72]). Gf is frequently assessed through tests such as Ravens matrices and the culture fair intelligence test ([22]), which are composed of figural reasoning tasks. Gc is typically measured through factual knowledge questions.

### 2.2. The Investment Trait Need for Cognition

Intellectual investment traits can be defined as “stable individual differences in the tendency to seek out, engage in, enjoy, and continuously pursue opportunities for effortful cognitive activity” ([69]). Consequently, investment traits may explain interindividual differences in the pursuit of learning opportunities, such as reading challenging literature, playing strategy games, or participating in political discussions. Thus, investment traits may enable individuals to create cognitively stimulating experiences promoting mental development and growth ([29]; [59]; [68]).

Investment traits encompass various psychological constructs such as openness to experience, vocational interests, and need for cognition (NFC), which is defined as an individual’s “tendency to engage in and enjoy thinking” ([14]). Different theoretical approaches exist for categorizing the differences and similarities between the various investment traits. According to [41] ([41]), the investment traits can be organized into a two-dimensional model that differentiates between two processes (Seek and Conquer) and three operations (Think, Learn, and Create). In this model, openness and NFC are categorized under Seek and Think, whereas interest falls under Seek and Learn. In a different approach, [68] ([68]) proposed that investment traits may be described by a hierarchical model. Here, NFC represents the general investment trait at the apex of the model as it refers to the core of investment: “the tendency to seek out, engage in, enjoy, and continuously pursue opportunities for effortful cognitive activity” ([69]). Most previous longitudinal research on the interplay between intelligence and investment traits included openness or interest. However, given the crucial role of NFC in both the theoretical frameworks of [41] ([41]) and [68] ([68]), we focused on this investment trait and its longitudinal interplay with intelligence in the present study.

According to [15] ([15]), high-NFC individuals are “chronic cognizers” (p. 197) with a tendency to delve deeply into problems, actively seek out new information, and reflect thoroughly on their discoveries ([46]). Theoretically and empirically, NFC has been associated with more profound and extensive information processing due to higher cognitive resource investment ([45]; [68]). This relationship is further supported by neurophysiological research (e.g., [41]; [61]). NFC does not systematically vary with gender, socioeconomic status, or cultural background ([4]; [6]; [20]). It is typically assessed through self-report measures. A recent meta-analysis found a modest average correlation (*r* = 0.20) between NFC and academic achievement, with this relationship becoming more pronounced as grade level increases ([36]). The incremental validity of NFC is demonstrated by its significant relationship with academic achievement even after controlling for cognitive ability, openness, persistence, academic self-concept, and intrinsic motivation ([31]; [62]). Students with high NFC have been reported to show greater engagement when anticipating challenging tasks ([57]), and NFC strongly predicts learning in cognitively demanding environments ([20]). Compared to other motivational variables such as achievement goals, academic interest, or academic self-concept, NFC is the most effective predictor of participation in gifted programs ([39]). These findings underscore the significance of NFC among all investment traits and suggest a central role in the developmental interplay between investment traits and intelligence.

### 2.3. Theories on the Development of Intelligence, Investment Traits, and Their Interplay

The investment theory, proposed by [16] ([16], [17]), is a well-established and influential framework describing the interplay between Gc, Gf, and investment traits. According to [17] ([17]), the observed cross-sectional correlation between Gc and Gf at any given time can be explained by the historical investment of Gf into the development of Gc. Thus, an individual’s current level of Gc is the cumulative result of several years of Gf investment. This investment, which also hinges on learning opportunities, occurs when Gf is utilized in various complex learning situations, leading to the acquisition of crystallized abilities, thus enabling individuals with higher Gf to accumulate knowledge more easily. Furthermore, the investment theory suggests that the development of Gc is supported by the time invested in learning, interests, and memory, which Cattell collectively referred to as the common learning investment (Se). Here, interest may be interpreted as an investment trait that leads to more learning opportunities and enables individuals to use their Gf to acquire Gc. Thus, interest, along with time and memory, is essential in directing Gf to develop Gc, explaining how individuals allocate their mental resources. [17] ([17]) discussed that the influences of Gf and Se could be either independent or interactive. An interaction would imply that developmental increases in Gc are particularly large when high Gf and Se occur together and larger than would be expected from the sum of the individual effects of Gf and Se.

In his intelligence-as-process, personality, interests, and intelligence-as-knowledge (PPIK) model, [1] ([1]) built upon Cattell’s investment theory. He differentiated between intelligence-as-process (Gf-type abilities) and intelligence-as-knowledge (Gc-type abilities). Ackerman assumed that a small set of personality factors are related to the development of intelligence-as-knowledge, particularly openness and different vocational interests (i.e., realistic, investigative, artistic, social, enterprising, and conventional interests). In line with Cattell’s investment theory, [1] ([1]) suggested that intelligence-as-process has a causal influence on intelligence-as-knowledge, such that individuals with higher fluid intelligence are likely to gather more knowledge. Additionally, the PPIK model posits that both intelligence-as-process and intelligence-as-knowledge positively influence investigative interests. Ackerman acknowledged that his theory was primarily based on cross-sectional correlations, as there were not many longitudinal studies available at the time to examine the longitudinal interplay between Gf, Gc, and investment traits. Nevertheless, cross-sectional research supports the positive relations between investment traits, Gf, and Gc ([2]; [13]; [52]; [68]).

[76] ([76]) built on investment theory and PPIK theory and added several extensions to these theories in their openness-fluid-crystallized-intelligence (OFCI) model. Like its predecessor theories, the OFCI model posits that Gf predicts changes in Gc over time, with a similar explanation as given by investment theory. [76] ([76]) highlighted openness to experience as the key investment trait, but without suggesting a direct path from previous openness to changes in Gc. Instead, their environmental enrichment hypothesis suggests that investment traits positively influence the development of Gf. According to this hypothesis, openness to experience encourages individuals to explore their environment, engage in social interactions, and immerse themselves in a stimulating and varied setting, which over time enhances their Gf development. As Gf predicts Gc development, investment traits indirectly affect Gc development through their impact on Gf. Additionally, according to their environmental success hypothesis, high Gf helps individuals master novel and unfamiliar challenges, which in turn fuels curiosity about new situations and promotes the positive development of investment traits. This results in gains in investment traits as individuals continually experience greater success due to higher Gf. Notably, unlike their predecessors, [76] ([76]) utilized longitudinal studies to test their hypotheses.

Finally, the Matthew effect and the compensation effect are key concepts in understanding how initial ability levels influence changes in ability over time. According to the Matthew Effect, higher initial ability levels facilitate the acquisition of new reasoning skills or knowledge over time ([56]). Consequently, individuals with already high Gf and Gc should experience the greatest gains, leading to a widening of the ability gap. In contrast, the compensation effect suggests that the gap between more and less able individuals decreases as less able individuals are able to catch up over time ([56]). In the context of education, the compensation effect could mean that schooling is particularly beneficial to lower-achieving students, promoting their cognitive development more effectively than that of higher-achieving students. In the same vein, a compensation effect may also arise from a lack of sufficient support for more able students ([9]; [56]).

### 2.4. Longitudinal Research on the Interplay Between Intelligence and Investment Traits

To obtain an overview of previous research findings, we conducted a literature search for studies that investigated the longitudinal relationships between Gc, Gf, and investment traits. We focused on studies that examined younger samples (i.e., children, adolescents, and young adults), as opposed to research on the effects of investment on cognitive aging (e.g., [38]; [44]; [67]; [75]). Furthermore, only studies that included longitudinal measures of all relevant constructs—Gc, Gf, and at least one investment trait—were included. The resulting eight studies are summarized in Table 1. When examining the samples, it was striking that almost all were from Germany, with one exception from China. The mean age at first assessment ranged from 8.4 to 17 years, broadly covering childhood and adolescence. Time intervals between the first and last assessments ranged from 0.5 to 6 years. To measure Gc, the studies used either Gc scores of cognitive ability tests (4 studies), achievement tests (3), or school exam scores (1), whereas Gf was always assessed by cognitive ability tests. The most frequently examined investment trait was openness (4 studies), followed by interest (3), achievement motivation and NFC (2 each), and curiosity and intellectual engagement (1 each).

The results regarding the prediction of change in Gc were heterogeneous. The prediction of change in Gc by earlier Gc was found to be either significantly positive ([3]; [74]) or negative ([5]; [26]; [56]; [76]). Gf was a significantly positive predictor for change in Gc in most studies ([3]; [32]; [76]), with the exception of [56] ([56]), who found a significantly negative effect. Investment traits predominantly showed no significant prediction for changes in Gc ([3]; [4]; [26]; [56]; [76]), with some exceptions of positive prediction ([32]; [74]). Finally, [32] ([32]) investigated the predictive effect of the interaction between Gf and investment traits, finding both nonsignificant (Gf × openness) and significantly positive (Gf × interest) prediction paths.

When predicting change in Gf, Gc was found to be either a nonsignificant predictor ([56]) or a positive predictor ([3]). For the prediction of change in Gf by earlier Gf, there was no clear pattern, with either nonsignificant ([26]), positive ([3]), or negative ([5]; [56]) effects. Lastly, investment traits were predominantly nonsignificant predictors of change in Gf ([3]; [5]; [26]; [56]). Only [76] ([76]) and [74] ([74]) reported a significant positive prediction. 

Finally, when predicting change in investment traits, Gc was found to be a nonsignificant predictor ([3]; [4]) or a positive predictor ([74]). [5] ([5]) found a positive prediction of change in investment traits only with mathematical Gc but not with verbal Gc. Prediction of change in investment traits by Gf was predominantly nonsignificant ([3]; [4]; [5]; [76]), with the exception of a positive prediction of the achievement motive “hope for success” ([4]; [5]). The prediction by earlier investment traits was either significantly positive ([3]; [74]) or negative ([5]; [76]).

Taken together, the longitudinal evidence for the predictions suggested by investment theory, PPIK theory, and the OFCI model is largely inconclusive. For most predictive paths, there was no convincing majority of studies suggesting a significant positive effect. For many paths, the evidence was very limited because all studies investigated only a selection of paths rather than all possible prediction paths. The large heterogeneity in sample age, test instruments, and investment traits examined complicates the identification of potential moderation patterns. 

### 2.5. Present Study

There are several theories on the longitudinal interplay between Gc, Gf, and investment traits. However, the relationships among these constructs have primarily been examined cross-sectionally, despite the inherently longitudinal nature of the research question. The few available longitudinal findings are inconclusive, and patterns are difficult to identify due to heterogeneous methods and investment trait measures. The fact that the available studies focused on different subsets of all possible longitudinal relationships further limits the informative value of this evidence. For instance, the prediction of change in investment traits by initial Gf was examined in only four of the reviewed longitudinal studies. Thus, previous research tended to focus on isolated pathways, leading to a fragmented understanding of the longitudinal interplay between these constructs. Moreover, previous longitudinal studies have examined a variety of investment traits such as openness, interest, intellectual engagement, and need for cognition, but the relationships remain inconsistent and heterogeneous. According to [68] ([68]), NFC is a highly general investment trait, making it an optimal candidate for testing the interplay between intelligence and investment traits. In the reviewed longitudinal studies, NFC was investigated twice ([4]; [5]). 

Our comprehensive overview of previous longitudinal studies highlights the inconsistency and heterogeneity of findings and underscores the need for a study that examines all possible pathways simultaneously. Accordingly, in our longitudinal study, we examine all possible reciprocal pathways between Gc, Gf, and NFC over time, providing a complete picture of relationships. In addition, we extend previous analyses by including the interaction between Gf and NFC, a relationship that has only been examined once before ([32]). Whenever there were clear theoretical expectations for a pathway, we formulated a specific hypothesis. This resulted in a total of ten hypotheses/research questions. The first four hypotheses/research questions refer to changes in Gc, followed by three hypotheses/research questions related to changes in Gf, and finally three hypotheses/research questions related to changes in NFC. In total, five directional hypotheses were stated.

**H1.** 
*T1 Gc positively predicts change in Gc. Directional hypothesis based on the Matthew Effect. Preregistered.*


**H2.** 
*T1 Gf positively predicts change in Gc. Directional hypothesis based on the investment theory. Preregistered.*


**H3.** 
*T1 NFC positively predicts change in Gc. Directional hypothesis based on the investment theory. Preregistered.*


***RQ4.*** 
*We investigate whether the interaction of T1 NFC and T1 Gf predicts change in Gc. Open research question. Preregistered.*


***RQ5.*** 
*We investigate whether T1 Gc predicts change in Gf. Open research question. Preregistered.*


**RQ6.** 
*We investigate whether T1 Gf positively predicts change in Gf. Open research question. Not preregistered.*


**H7.** 
*T1 NFC positively predicts change in Gf. Directional hypothesis based on the environmental enrichment hypothesis of the OFCI model. Preregistered.*


**RQ8.** 
*We investigate whether T1 Gc predicts change in NFC. Open research question. Preregistered.*


**H9.** 
*T1 Gf positively predicts change in NFC. Directional hypothesis based on the environmental success hypothesis of the OFCI model. Preregistered.*


**RQ10.** 
*We investigate whether T1 NFC positively predicts change in NFC. Open research question. Not preregistered.*


Note that in our preregistration (Accessed on 20 October 2024: https://doi.org/10.17605/OSF.IO/9547T), we did not explicitly distinguish between directional hypotheses and open research questions. We also used a different organizational structure, although the content of the hypotheses/research questions remained the same. RQ6 and RQ10 were added post hoc.

## 3. Methods

The analysis was preregistered prior to data collection (Accessed on 20 October 2024: https://doi.org/10.17605/OSF.IO/9547T). Note that our preregistration includes interest as a second investment trait alongside NFC. In the present study, we focus on NFC because it is a domain-general representation of investment traits. According to [68] ([68]), NFC can be interpreted as the general factor of different investment traits, making it an optimal candidate for examining the interplay between intelligence (i.e., Gf and Gc assessed as broad constructs in our study) and investment traits. In contrast, academic interests were assessed as domain-specific constructs. Based on Brunswik symmetry ([12]; see also [30]), we decided not to include them in this study. However, we plan to explore interest in a separate study with domain-specific outcomes (academic achievement in the same domains for which interest was assessed).

### 3.1. Participants and Procedure

We assessed 424 students (Grades 7–9) from four schools at the beginning of the school year in late August or early September 2020 and 387 students at the end of the same school year (10-month interval) in June or July 2021. The students were enrolled in either regular classes or special classes for the gifted (45.75%) across 36 different classes at four German grammar schools in Rhineland-Palatine in Germany. Grammar schools form the highest track of the German three-track secondary school system. 341 students participated at both measurement points. This overlapping sample is the basis of this study. The mean age at T1 was *M* = 13.35 years (*SD* = 0.94). 194 students (56.89%) identified as male, 139 (40.76%) as female, and 8 (2.35%) as nonbinary.

The testing was not part of the school curriculum and carried out exclusively for research purposes. Students received written personalized feedback on their ipsative intelligence and vocational interest profiles at the end of the study. In addition, schools received a report summarizing the aggregated results of their students. Students were tested in groups in classroom settings. The group size during testing ranged from 2 to 26 students (*M* = 12.73, *SD* = 6.61). At both measurement points, the testing session lasted approximately five hours in total. First, students’ intelligence was assessed using the BIS-HB (Berlin Structure-of-Intelligence Test: Assessment of Giftedness and Talent; [28]). During this three-hour assessment, two 15-min breaks were taken. After an additional 15-min break, crystallized intelligence was measured for 30 min using the Gc scale of the Berlin Test for the Assessment of Fluid and Crystallized Intelligence for Grades 8 to 10 (BEFKI 8–10; [72]). Following another 15-min break, demographic information, need for cognition, and further motivational variables were collected through self-reports. Trained student assistants (undergraduates) and the principal investigators administered the tests, with two administrators present per group. All parents of participants provided written informed consent in accordance with the Declaration of Helsinki. The protocol was approved by the principals of the participating schools. The data collection was approved by the Supervision and Service Directorate of Rhineland–Palatinate on the basis of ethical and data protection requirements (Aufsichts- und Dienstleistungsdirektion; protocol numbers 153-20 and 226-21).

### 3.2. Materials

#### 3.2.1. Crystallized Intelligence

Gc was assessed with the Gc scale from the BEFKI 8–10 ([72]). The Gc scale contains a total of 64 items and measures declarative knowledge in 16 areas. The 16 areas of knowledge can be categorized into the three major domains: natural sciences, humanities, and social sciences. Sample reliability was α/ω = 0.84/0.84 at T1 and α/ω = 0.84/0.84 at T2.

#### 3.2.2. Fluid Intelligence

Multiple facets of student intelligence were assessed with the BIS-HB ([28]). The BIS-HB is a paper-and-pencil cognitive ability test designed to assess cognitive ability structure across the full range of abilities, with a particular focus on adequately testing intellectually gifted students. The BIS-HB is based on [27]’s ([27]) Berlin Model of Intelligence Structure (BIS), which is a two-faceted model of intelligence that consists of an operation facet and a content facet. The operation facet includes processing speed, memory, creativity, and reasoning. The content facet includes verbal, numerical, and figural abilities. Each individual test item is assigned to a combination of one operation and one content (e.g., a figural memory task). 

To represent Gf as closely as possible to the way it is discussed and rationalized in investment theory ([16], [17]), we chose the figural reasoning subscale of the BIS-HB. Compared to verbal or numerical reasoning subscales, figural reasoning indicators are often considered more independent from cultural and educational influences, although there is a consensus that Gf can never be entirely free from such influences (e.g., [19]; [23]). The figural reasoning subscale of the BIS-HB is assessed by five different tasks (i.e., figural analogies, Charkow problems, Bongard problems, figure selection, and mental folding tasks). Sample reliability was α/ω = 0.68/0.71 at T1 and α/ω = 0.72/0.75 at T2.

#### 3.2.3. Need for Cognition

Need for cognition (NFC) was assessed with the German version of the NFC-KIDS scale (14 positively worded items; e.g., “Thinking is fun for me”; [49]). Reliability was α/ω = 0.89/0.90 at T1 and α/ω = 0.91/0.92 at T2.

### 3.3. Data Analyses

Analysis code is available online on: https://doi.org/10.17605/OSF.IO/9547T (Accessed on 20 October 2024). The data analyses were conducted with R 4.4.1 and Mplus 8.7. In all Mplus analyses, we used the maximum likelihood estimator with robust standard errors. To account for the nested data structure (i.e., students nested within classes), we applied the “type is complex” option in all Mplus analyses. Missings were accounted for by the FIML algorithm.

#### 3.3.1. Measurement Invariance over Time

Before testing our hypotheses, we tested whether Gc, Gf, and NFC were measurement invariant over the two time points. To test the measurement invariance over time, we conducted several autocorrelative structural equation models (SEMs) and successively tested increasing measurement invariance levels against each other (i.e., configural, metric, and scalar). We applied effect coding to identify the models ([35]). Following [18] ([18]), we used comparative fit indices (ΔCFI) to compare two models with different levels of measurement invariance. ΔCFI values of 0.01 or less were interpreted as indicating a tolerable deterioration in model fit.

#### 3.3.2. Latent Change Score Analyses

We used latent change score (LCS) models to test whether the initial levels of Gc, Gf, and NFC (i.e., the T1 measure) predicted the latent change score of the other variables. First, to test whether there is variation in the change of Gc, Gf, and NFC, we estimated separate LCS for each variable. We then followed a three-step approach that included multiple variables in each model. In Step 1, the LCSs of any two constructs were observed simultaneously (i.e., NFC and Gf; NFC and Gc; Gf and Gc). In Step 2, we applied one model containing all three variables. In this model, the effect from one variable on the latent change score of another variable was statistically controlled by the third variable (e.g., the effects from T1 NFC on the LCS of Gc were statistically controlled by T1 Gf). In Step 3, the model from Step 2 was extended to include the latent interaction score of Gf and NFC at T1, which predicted the LCS of Gc. 

#### 3.3.3. Robustness Analyses

The students attended either regular classes or special classes for the gifted. To address this potential confound, we conducted a robustness analysis that included class type (i.e., regular vs. gifted) as an additional predictor of LCS for Gc, Gf, and NFC in Steps 1 through 3 of our analysis. We decided not to include additional control variables in our model, as this may lead to unintended variance restrictions. The aim of our study was to examine the relationships between Gc, Gf, and NFC both independently and while controlling for the third variable. Because we did not aim to establish causal effects, we did not include additional control variables, which means that alternative explanations for the observed relationships remain possible.

## 4. Results

Latent correlations and descriptive statistics are reported in Table 2. All constructs were significantly related to all other constructs at all measurement points (*r* range = 0.22 to 0.65, *p* < .05), with the exception of Gf_T1 and NFC_T1 with an only marginally significant correlation (*r* = 0.17, *p* = .056). The autocorrelations of Gc (*r* = 0.92, *p* < .001) and Gf (*r* = 0.94, *p* < .001) were very high, while the autocorrelation of NFC was slightly lower (*r* = 0.73, *p* < .001). On average, Gc (*d*_T1-T2_ = 0.15) and Gf (*d*_T1-T2_ = 0.20) slightly increased from T1 to T2, while NFC (*d*_T1-T2_ = −0.04) showed no substantial mean-level change over time. The intraclass correlation coefficients (ICCs) for all variables were substantial (ICC > 0.10), particularly for Gc (T1: ICC = 0.44; T2: ICC = 0.47), indicating that there are larger differences in Gc between classes, possibly because we examined both regular classes and classes for gifted. Manifest correlations and descriptive statistics in regular classes and special classes for gifted are reported in Appendix A. The results for regular classes and special classes for gifted students were largely comparable, with one minor exception. While NFC decreased slightly in regular classes (*d*_T1-T2_ = −0.27), there was no relevant change in the gifted classes (*d*_T1-T2_ = −0.04).

### 4.1. Measurement Invariance over Time

In Gc and Gf, we found scalar measurement invariance over the measurement points (i.e., ΔCFI < 0.01 between models of different measurement invariance levels; see Table 3), allowing for mean-level comparisons over time. In NFC, we achieved partial scalar measurement invariance across measurement points by allowing the intercept of Item 5 to vary across measurement points.

### 4.2. Latent Change Score Analyses

Separate LCS analyses for Gc, Gf, and NFC revealed significant increases in Gc (LCS Mean = 1.28, *p* < .001) and Gf (LCS Mean = 4.09, *p* < .001), but no significant change in NFC (LCS Mean = −0.03, *p* = .495). There was significant variation in the LCS for each variable (Gc: Variance = 1.15, *p* = .001; Gf: Variance = 8.37, *p* = .035; NFC: Variance = 0.23, *p* < .001). The results of latent change score analyses in Step 1, Step 2, and Step 3 are reported in Figure 1, Figure 2, and Figure 3, respectively. In Step 1, the LCS of Gc was negatively predicted by the initial Gc, with β = −0.38 (*p* = .008) when controlling for the initial Gf, and β = −0.31 (*p* = .001) when controlling for the initial NFC. The LCS of Gf was positively predicted by the initial NFC, with β = 0.36 (*p* = .008). The LCS of NFC was positively predicted by both the initial Gc (β = 0.26, *p* < .001) and Gf (β = 0.29, *p* < .001) and negatively predicted by the initial NFC, with β = −0.25 (*p* < .001) when controlling for the initial Gc, and β = −0.26 (*p* < .001) when controlling for the initial Gf.

In Step 2, the LCS of Gc was negatively predicted by the initial Gc, with β = −0.42 (*p* < .001). The LCS of Gf was positively predicted by the initial NFC, with β = 0.38 (*p* = .007). The LCS of NFC was positively predicted by the initial Gf (β = 0.22, *p* < .001) and negatively predicted by the initial NFC, with β = −0.26 (*p* < .001). 

In Step 3, the interaction between the initial Gf and the initial NFC did not significantly predict the LCS of Gc (β = −0.16, *p* = .227). All significant prediction paths from Step 2 remained significant.

The results of the robustness analyses for Steps 1 to 3 are reported in Appendix A. The findings remained largely unchanged when class type was included in the model. Specifically, the significant paths remained significant, and the non-significant paths remained non-significant.

## 5. Discussion

Previous findings on the longitudinal interplay between intelligence and investment traits are inconclusive. We conducted a longitudinal study in four German high-track secondary schools, assessing 341 students twice over one year. Gc and Gf were measured as indicators of intelligence, while NFC was assessed as a measure of investment traits. Results revealed scalar measurement invariance over time for each measure, indicating that the constructs were assessed consistently over time, allowing for the interpretation of mean-level changes. All possible longitudinal relationships between the three constructs were examined using a stepwise approach. A summary of the results is presented in Table 4. LCS analyses revealed that change in Gc over time was negatively predicted by initial Gc and not significantly predicted by initial Gf or initial NFC, contrary to theoretical assumptions. Changes in Gf over time were significantly and positively predicted by initial NFC but not by initial Gc or Gf. Changes in NFC over time were positively predicted by initial Gc (but not when controlling for initial Gf), positively predicted by initial Gf, and negatively predicted by initial NFC.

### 5.1. Predicting Change in Gc, Gf, and NFC

**H1**: Contrary to our hypothesis, initial Gc negatively predicted changes in Gc. A positive prediction, which was observed in some previous longitudinal studies ([3]; [74]), would be plausible according to the Matthew effect, where higher initial Gc facilitates further development ([56]). However, a negative prediction is not an unusual result and has also been observed in other previous longitudinal studies ([5]; [26]; [56]; [76]). This negative prediction path can be explained by the Compensation Effect, which describes that weaker students catch up over time and compensate for their initially lower levels. Alternatively, the negative prediction path in our study may be explained by regression to the mean, where extreme initial scores tend to move towards the average over time.

**H2**: Contrary to our hypothesis and the theoretical expectations of the investment theory, PPIK model, and OFCI model, changes in Gc were not significantly predicted by initial Gf, although the relationship was descriptively positive. This nonsignificant effect is surprising because many previous longitudinal studies reported a significant positive effect of initial Gf ([32]; [76]) on Gc development, although not consistently ([3]; [56]). Studies with significant positive effects often used longer intervals (i.e., two years and longer), suggesting that the effect might be too small to detect with shorter test-retest intervals. Thus, it is possible that a longer time frame is required to observe a substantial influence of Gf on changes in Gc.

**H3**: Initial NFC also did not significantly predict changes in Gc, contrary to our hypothesis, although the relationship was descriptively positive. Similar to H2, it is possible that the time frame was too short to observe meaningful changes in Gc. However, it is also important to note that previous longitudinal findings on this relationship are quite inconsistent (see Table 1). Furthermore, while the investment theory and PPIK model suggest that this relationship is plausible, the OFCI model does not assume a direct path from earlier investment traits to changes in Gc. Instead, the OFCI model posits that investment traits indirectly affect Gc through changes in Gf.

**RQ4**: The interaction between initial Gf and initial NFC did not predict changes in Gc. [17] ([17]) discussed whether the effects of Gf and investment traits on Gc are additive (no interaction) or multiplicative (with interaction). Such an longitudinal interaction has only been investigated once before, where no significant interaction was found for interests as an investment trait, but a significant interaction was found for openness as an investment trait ([32]). Given that the main effects of both initial Gf and initial NFC were not significant predictors in our study, it is not surprising that their interaction was also not significant in these data. As in H2 and H3, it is possible that one year is not sufficient to observe a significant change in Gc. Nevertheless, our current study does not support the hypothesis of multiplicative effects of Gf and NFC on changes in Gc.

**RQ5**: We investigated whether initial Gc predicts changes in Gf as an open research question. This prediction path is typically not considered by the relevant theories, as it is generally hypothesized that Gf is the antecedent and Gc the product. In previous longitudinal studies, [3] ([3]) found a positive effect, whereas [56] ([56]) did not observe a significant effect. Our data do not provide evidence that Gc is the antecedent of Gf; however, as stated in the discussion of H2, we did not observe the opposite effect in our data either.

**RQ6**: LCS analyses indicated neither a significant positive nor a negative prediction path from initial Gf on changes in Gf. Our data therefore support neither the Matthew Effect nor the Compensation Effect. Findings in the previous longitudinal research on this relationship are heterogeneous (see Table 1). 

**H7**: In line with our hypothesis, NFC positively predicted changes in Gf. Unlike previous theories, the OFCI model proposes this pathway based on the environmental enrichment hypothesis. Previous longitudinal studies have sometimes found this relationship ([74]; [76]), although many other longitudinal studies have reported null findings (e.g., [3]; [26]). Our study supports the notion that high NFC promotes Gf development, potentially, as proposed by the environmental enrichment hypothesis, because NFC motivates individuals to explore their environment, engage in social interactions, and immerse themselves in a stimulating and varied environment.

**RQ8**: Initial Gc positively predicted changes in NFC. This effect disappeared when initial Gf was included as an additional predictor in the model. This suggests that the shared variance between Gf and Gc is responsible for the effect rather than the initial Gc alone. The role of Gf in the relationship is consistent with the environmental success hypothesis, which posits that prior intellectual success in novel and unfamiliar challenges fuels the enjoyment of thinking and promotes the positive development of investment traits. The pathway from initial Gc on changes in investment traits is typically not considered in the prominent theories. Previous longitudinal research on this relationship has been inconclusive, showing both positive predictions ([74]) and null findings ([3]; [4]). Our results suggest a new perspective, implying that Gf may potentially play a mediating role in the relationship between initial Gc and changes in NFC. Further research, such as longitudinal mediation analyses, is needed to better understand this relationship. 

**H9**: Consistent with our expectations and the OFCI model, initial Gf predicted changes in NFC. Previous longitudinal literature on this relationship has mostly reported null findings ([3]; [76]), with some exceptions ([4]; [5]). Our findings can be explained by the environmental success hypothesis and support the notion that intellectual success at an earlier stage promotes the positive development of NFC over time.

**RQ10**: Initial NFC negatively predicted changes in NFC over time. Similar to H1, this result could be explained by methodological artifacts such as regression to the mean.

### 5.2. Limitations

Before discussing the implications, we note the limitations of our study. In our literature overview, we highlighted that previous longitudinal research has been conducted almost exclusively in Germany, which limits the generalizability of the findings beyond this population. Due to the design of the original data collection, we could not address this limitation of the broader field in our study, as our data were also collected in Germany. In addition, our study only focused on one type of adolescent school experience, that of German students in the highest track of the secondary school system. Future studies should address this limitation by conducting comparable analyses in other countries and school forms. Another limitation is that students either attended regular classes or gifted classes within this school type, which could potentially confound the longitudinal interplay between Gc, Gf, and NFC. To address this, we conducted a robustness analysis including class type as an additional predictor, and the results remained largely unchanged. In addition, we decided not to include other control variables in our model. As a result, we cannot rule out that third variables explain the observed longitudinal relationships.

A second limitation is that our longitudinal study spanned only one school year. This time interval might be insufficient for substantial changes in Gc or Gf to occur in most students. According to the recent meta-analysis by [10] ([10]), the expected one-year autocorrelations in fourteen-year-old adolescents for Gc and Gf are 0.92 and 0.87, respectively, when controlling for the reliability of the tests. In our models for testing measurement invariance, we found latent autocorrelations of 0.92, 0.94, and 0.73 for Gc, Gf, and NFC, respectively. Thus, many participants maintained their relative rank, particularly in Gc and Gf, which minimizes the variance of interindividual differences in intraindividual development in these constructs. This underlines the importance of our finding that, despite the high stability of Gf, NFC can still predict changes in Gf. In addition, it can also be argued that one year is a significant period. For instance, [17] ([17]) proposed that “the last year will be most important—in the case of growing children, but not adults—because the fluid ability a year earlier will not have been at a high enough level to account for the summit level of this year’s crystallized intelligence.” Students also learn a substantial amount of new knowledge over one school year, leading to significant changes in Gc and differences in the rates of these changes. Accordingly, our analyses revealed a significant variation in the latent change scores of Gc, Gf, and NFC. 

A similar limitation is that our study included only one follow-up assessment, which limits the analyses in several ways. With only two measurement occasions, it is difficult to distinguish between true change and regression to the mean, which may explain why initial Gc scores negatively predicted changes in Gc. Having at least three measurement occasions would allow us to better distinguish between true developmental trends and statistical artifacts. It would also allow for more accurate modeling of growth parameters ([48]) and better examination of nonlinear developmental effects. In addition, multiple measurement occasions over several years would help to examine the impact of test-retest intervals on the prediction of change in intelligence and investment traits and would allow investigating possible long-term mediation effects. 

Although we examined a relatively large sample of 341 students who participated at both measurement points, the sample size might still be too small to detect small effects. For example, the prediction of changes in Gc by initial NFC was not significant, but the estimates and *p*-values were β = 0.25 (*p* = .065) in Step 1 and β = 0.24 (*p* = .082) in Step 2 of our analyses. With a larger sample size and greater statistical power, this effect might have become significant. In addition, when investigating the latent interaction between Gf and NFC in Step 3 of our analysis, we encountered several Heywood cases, which may also be due to the limited sample size for these complex analyses.

Furthermore, it is important to highlight that, although our study design is longitudinal and thus provides more insight into changes over time than cross-sectional research, the LCS analyses are still based on correlations and do not test for causal effects. Finally, our study was conducted during the COVID-19 pandemic. Different forms of schooling, such as home schooling or hybrid attendance, may have influenced the way in which new knowledge or skills were acquired ([11]). As a result, our findings may not fully represent what would happen during a regular school phase without interruptions.

### 5.3. Implications

Our study found significant longitudinal relationships between Gf and NFC but hardly any relationship between Gc and either GF or NFC. Different models of investment traits assume different roles of NFC. [41] ([41]) categorized NFC under the Think operation, meaning that NFC specifically affects motivation related to thinking and problem solving but not to learning or creating. Because we found that NFC was related to Gf, which relates to the Think operation, but not to Gc, which more closely relates to the Learn operation, our findings support Mussel’s assumption of a think-specificity of NFC. However, it is important to note that our study was not designed to explicitly test these theoretical assumptions.

Regarding the theories on the interplay between Gc, Gf, and investment traits, our data did not support the core assumptions of the investment theory, the PPIK model, or the PFCI model, which all postulate that the development of Gc can be explained by previous Gf and previous investment traits. However, our findings are in line with both the environmental enrichment hypothesis and the environmental success hypothesis included in the OFCI model, suggesting that Gf and investment traits are important for each other’s development.

Our present study and the review of other longitudinal studies lead to the following implications for future research: First, more than two measurement points are desirable, as this allows to test not only reciprocal effects but also potential mediations, such as the effect of investment traits on Gc mediated by Gf, as proposed by the PFCI model. In addition, it is important to consider the length of time between measurements to ensure that significant changes in Gf and Gc can occur. A period of just under a year, as in our study, may be too short, and the rank-order stability of Gf and Gc during this period may be too high. According to the meta-analysis by [10] ([10]), the stability coefficients for Gc and Gf in 14-year-olds, adjusted for test reliability, are 0.92 and 0.87 over one year, 0.91 and 0.85 over two years, and 0.88 and 0.82 over five years, respectively. Furthermore, we strongly encourage researchers to conduct similar studies outside of Germany, as all longitudinal studies on this topic, with one exception, have been conducted exclusively in Germany. Finally, although the prediction of Gf on changes in Gc is theoretically plausible, it is surprising that this relationship is not consistently observed in the literature and was not evident in our study. Future research may need to further explore this relationship, taking into account possible moderators such as learning opportunities, SES, and investment traits—factors already theorized in Cattell’s investment theory.

## Figures and Tables

**Figure 1 jintelligence-12-00104-f001:**
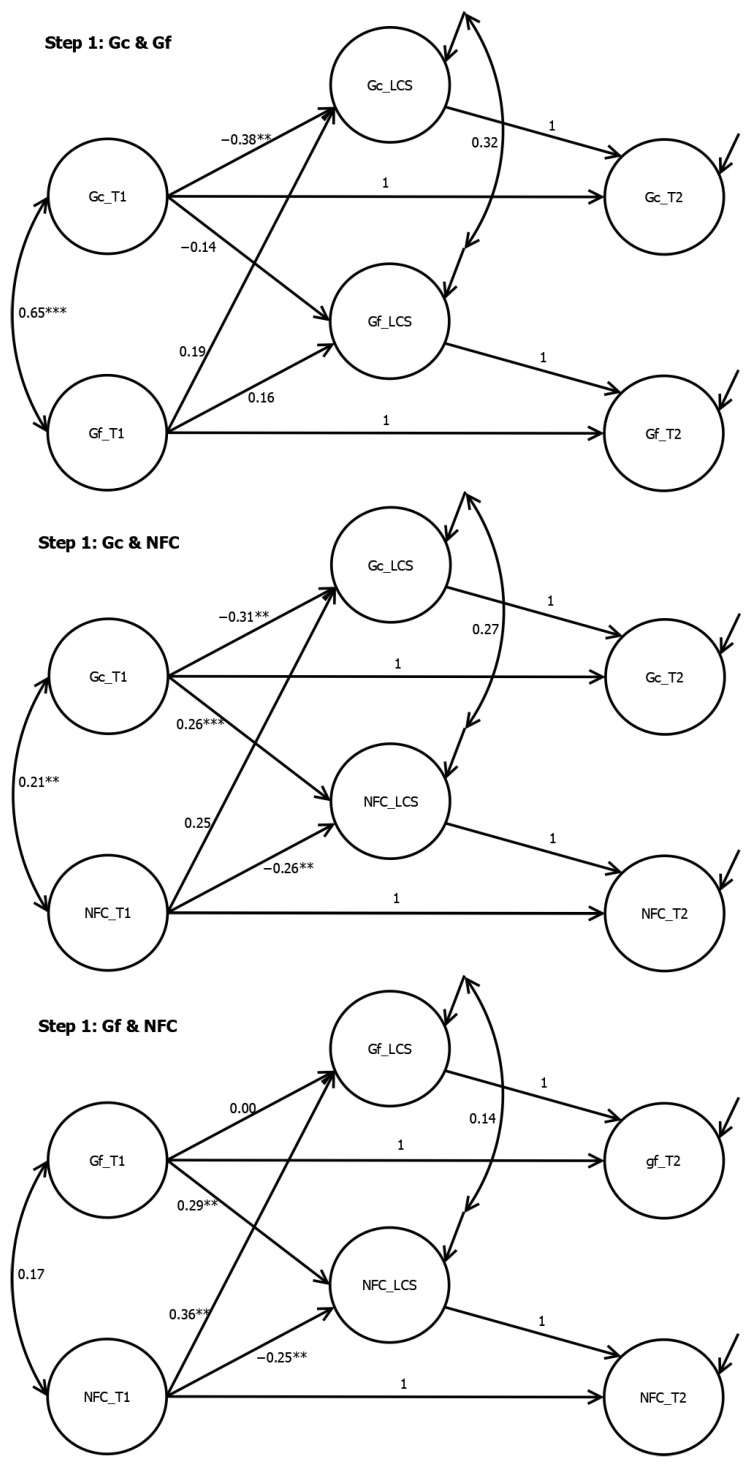
Bidirectional Latent Change Score Analyses from Step 1. Note. *** < .001. ** < .01.

**Figure 2 jintelligence-12-00104-f002:**
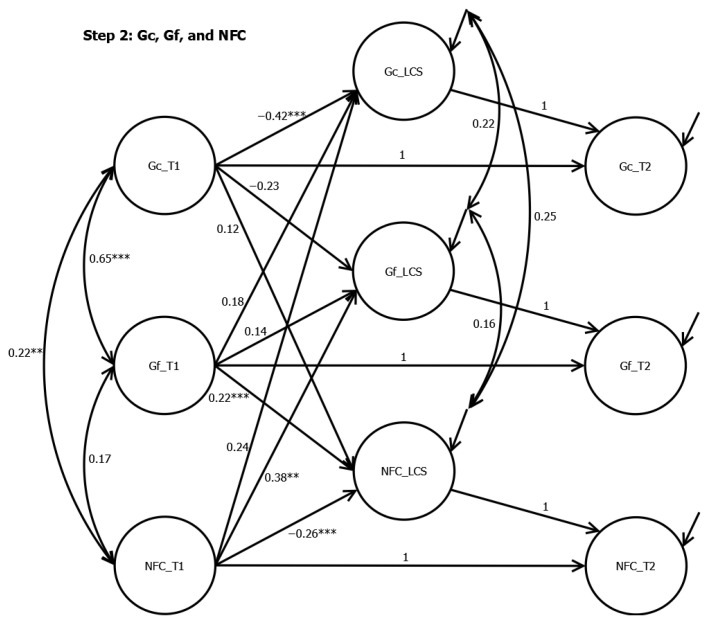
Latent Change Score Analyses from Step 2. Note. *** < .001. ** < .01. NFC = Need for cognition. Gf = Fluid intelligence. Gc = crystallized intelligence.

**Figure 3 jintelligence-12-00104-f003:**
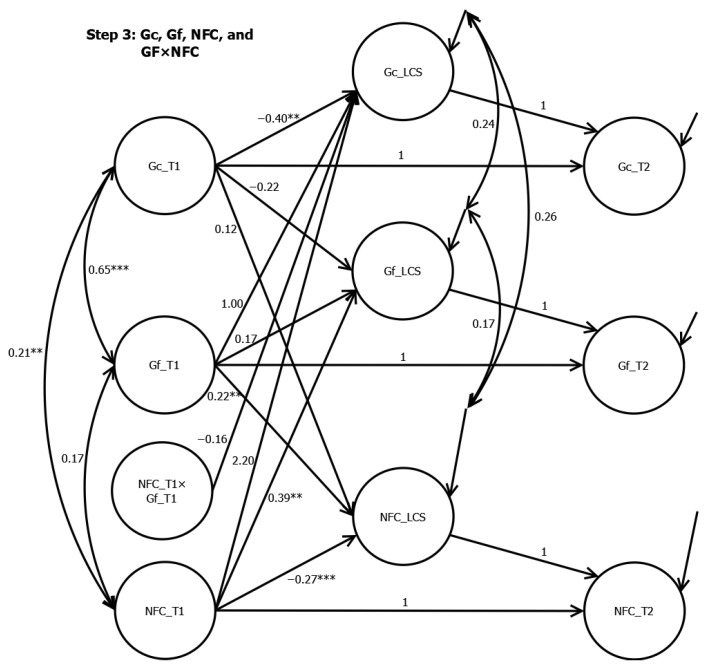
Latent Change Score Analyses from Step 3. Note. *** < .001. ** < .01. NFC= Need for cognition. Gf = Fluid intelligence. Gc = crystallized intelligence. Although the model estimation terminated normally according to Mplus, two Heywood cases appeared after the inclusion of the interaction term between Gf_t1 and NFC_t1. The prediction paths from Gf_t1 (β = 1.00) and NFC_T1 (β = 2.20) to Gc_LCS were greater than 1, although they were not significant.

**Table 1 jintelligence-12-00104-t001:** Longitudinal Research on the Interplay between Gc, Gf, and Investment Traits.

Reference	Sample	Country	Measurements	Method	H1 Gc → ΔGc	H2 Gf → ΔGc	H3 Inv → ΔGc	RQ4 Gf × Inv → ΔGc	RQ5 Gc → ΔGf	RQ6 Gf → ΔGf	H7 Inv → ΔGf	RQ8 Gc → ΔInv	H9 Gf → ΔInv	RQ10 Inv → ΔInv
[3] ([3])	*N* = 3880 T1: age 11 T2: age 12 T3: age 13 T4: age 14 T1–T2 ^1^ T1–T3 ^2^ T1–T4 ^3^	Germany	**Gc:** Math Achievement Test **Gf:** KFT 4–12+ **Investment:** Openness	Cross-Lagged Effects SEM	+ ^1^ + ^2^ + ^3^	0 ^1^ + ^2^ 0 ^3^	0 ^1^ 0 ^2^ 0 ^3^		+ ^1^ + ^2^ + ^3^	+ ^1^ 0 ^2^ + ^3^	0 ^1^ 0 ^2^ 0 ^3^	0 ^1^ 0 ^2^ 0 ^3^	0 ^1^ 0 ^2^ 0 ^3^	+ ^1^ + ^2^ + ^3^
[5] ([5])	*N* = 445 T1: age 8.4 T2: age 9.4	Germany	**Gc:** ELFE II—Text Comprehension ^1^, HRT 1–4—Arithmetic Operations ^2^ **Gf:** CFT 20-R **Investment:** NFC ^a^; Hope for Success ^b^; Fear of Failure ^c^; I-Type Curiosity ^d^; D-Type Curiosity ^e^	Latent Change Score SEM	− ^1^ − ^2^		+ ^1a^ 0 ^1b^ 0 ^1c^ 0 ^1d^ 0 ^1e^ 0 ^2a^ 0 ^2b^ 0 ^2c^ − ^2d^ 0 ^2e^			−	0 ^a^ 0 ^b^ 0 ^c^ 0 ^f^ 0 ^e^	0 ^1a^ 0 ^1b^ 0 ^1c^ 0 ^1d^ 0 ^1e^ + ^2a^ + ^2b^ + ^2c^ + ^2d^ 0 ^2e^	0 ^a^ + ^b^ 0 ^c^ 0 ^d^ 0 ^e^	− ^a^ − ^b^ − ^c^ − ^d^ − ^e^
[4] ([4])	*N =* 476 T1: age 16.43 T2: age 16.93 T3: age 17.43 T4: age 18.43	Germany	**Gc:** IST 2000-R—Gc **Gf:** IST 2000-R—Gf **Investment:** Hope for Success ^1^, Fear of Failure ^2^, NFC ^3^	Second-Order Latent Growth SEM								0 ^1^ 0 ^2^ 0 ^3^	+ ^1^ 0 ^2^ 0 ^3^	
[26] ([26])	*n* = 112 T1 = age 14.7 T2 = age 16.7	Germany	**Gc:** BEFKI—Gc **Gf:** BEFKI—Gf **Investment:** Intellectual Engagement	Latent Change Score SEM	−		0			0	0			
[32] ([32])	*n* = 4646 T1 = Grade 5–7 T2 = Grade 9	Germany	**Gc:** Reading Competence ^1^; Math Competence ^2^ **Gf:** NEPS matrices test **Investment:** Openness ^a^; Interest in Reading ^b^; Interest in Math ^c^	Latent Change Score SEM		+ ^1^ + ^2^	0 ^1a^ + ^1b^ 0 ^2a^ + ^2c^	0 ^1a^ + ^1b^ 0 ^2a^ + ^2c^						
[56] ([56])	*N =* 1102 T1: Grade 3 start T2: Grade 3 middle term	Germany	**Gc:** BEFKI—Gc **Gf:** BEFKI—Gf **Investment:** Math Interest ^a^; German Interest ^b^; Math Self-concept ^c^; German Self-concept ^d^	Latent Change Score SEM	−	−	0 ^a^ 0 ^b^ 0 ^c^ 0 ^d^		0	−	0 ^a^ 0 ^b^ + ^c^ 0 ^d^			
[74] ([74])	*N =* 823 T1: age 15.23 T2: age 15.73 T3: age 16.23 T1–T2 ^1^ T2–T3 ^2^	China	**Gc:** Exam scores **Gf:** Raven SPM **Investment:** Openness ^a^; Interest ^b^ English Model ^x^ Math Model ^y^	Cross-Lagged Effects SEM	+ ^1x^ + ^1y^ + ^2x^ + ^2y^		0 ^1ax^ + ^1ay^ + ^1bx^ + ^1by^ 0 ^2ax^ + ^2ay^ + ^2bx^ + ^2by^				+ ^1ax^ + ^1ay^ 0 ^1bx^ 0 ^1by^ + ^2ax^ + ^2ay^ 0 ^2bx^ 0 ^2by^	+ ^1ax^ + ^1ay^ + ^1bx^ + ^1by^ + ^2ax^ + ^2ay^ + ^2bx^ + ^2by^		+ ^1ax^ + ^1ay^ + ^1bx^ + ^1by^ + ^2ax^ + ^2ay^ + ^2bx^ + ^2by^
[76] ([76])	*n* = 172 T1 = age 17 T2 = age 23	Germany	**Gc:** HAWIE-R Vocabulary **Gf:** CFT **Investment:** Openness (parent rating)	Latent Change Score SEM	−	+	0			−	+		0	−

Note. + indicates that a significant positive prediction path was observed. 0 indicates that the prediction was investigated but no significant path was found. − indicates that a significant negative prediction path was observed. BEFKI = Berlin test for the assessment of fluid and crystallized intelligence for grades 8 to 10 (Berliner Test zur Erfassung fluider und kristalliner Intelligenz für die 8. bis 10. Jahrgangsstufe; [72]). ELFE II = A reading comprehension test for first to seventh graders—Version II (Ein Leseverständnistest für Erst- bis Siebtklässler—Version II; [33]). HAWIE-R = Hamburg-Wechsler intelligence test for adults—revision 1991 (Hamburg–Wechsler Intelligenztest für Erwachsene—Revision 1991; [63]). IST 2000-R = Intelligence Structure Test 2000 R (Intelligenz-Struktur-Test 2000 R; [34]). I-Type Curiosity = curiosity as a feeling of interest. D-Type Curiosity = curiosity as a feeling of deprivation. KFT 4–12+ = Cognitive ability test for grades 4 to 12 (Kognitiver Fähigkeitstest für 4. bis 12. Klassen; [25]). NFC = Need for cognition. Raven SPM = Raven’s Standard Progressive Matrices ([50]).

**Table 2 jintelligence-12-00104-t002:** Latent Correlations and Descriptive Statistics of the Investigated Variables.

	ICC	*M*	*SD*	*d* _T1-T2_	Gc_T1	Gc_T2	Gf_T1	Gf_T2	NFC_T1
Gc_T1	0.44	11.92	8.31	0.15					
Gc_T2	0.47	13.20	8.31		0.92 ***				
Gf_T1	0.24	101.46	20.31	0.20	0.65 ***	0.64 ***			
Gf_T2	0.24	105.54	20.50		0.58 ***	0.62 ***	0.94 ***		
NFC_T1	0.10	3.29	1.11	−0.04	0.22 **	0.29 ***	0.17	0.28 ***	
NFC_T2	0.12	3.25	1.29		0.33 ***	0.44 ***	0.33 ***	0.42 ***	0.73 ***

Note. *** < .001. ** < .01. Correlation parameters estimated based on a CFA modeling Gc_T1, Gc_T2, Gf_T1, Gf_T2, NFC_T1, and NFC_T2 as six correlated factors. This CFA reached an acceptable model fit (CFI = 0.920, RMSEA = 0.048, SRMR = 0.075). *d* $=\frac{MT2-MT1}{SDT1}$. The ICC represents the proportion of variance in a variable that is due to differences between school classes compared to the total variance.

**Table 3 jintelligence-12-00104-t003:** Model Fit Parameters for all Conducted Models.

Model	*χ*²	*df*	SCF	*p*	*CFI*	*RMSEA*	*SRMR*	Δ*χ*²	Δ*df*	Δ*p*	Δ*CFI*
**Latent Correlations Model**											
Correlations	1512.975	848	1.009	<.001	0.920	0.048	0.075				
**Measurement Invariance Tests**											
Gc configural	9.797	5	0.967	.081	0.996	0.053	0.012				
Gc metric	12.791	7	0.951	.077	0.995	0.049	0.029	2.953	2	.228	−0.001
Gc scalar	12.201	9	1.183	.202	0.997	0.032	0.024	1.138	2	.566	0.002
Gf configural	26.315	29	1.113	.609	1.000	0.000	0.023				
Gf metric	28.590	33	1.148	.686	1.000	0.000	0.042	2.520	4.000	.641	0.000
Gf scalar	35.216	37	1.143	.553	1.000	0.000	0.048	6.744	4.000	.150	0.000
NFC configural	731.068	281	1.052	<.001	0.911	0.069	0.073				
NFC metric	755.068	294	1.047	<.001	0.909	0.068	0.075	22.869	13.000	.043	−0.002
NFC scalar	822.068	307	1.049	<.001	0.898	0.070	0.077	65.611	13.000	<.001	−0.011
NFC partial scalar	792.708	306	1.047	<.001	0.904	0.068	0.076	37.640	12.000	<.001	−0.005
**Latent Change Score Analyses**											
Gc only	12.201	9	1.183	.202	0.997	0.032	0.024				
Gf only	35.216	37	1.143	.553	1.000	0.000	0.048				
NFC only	792.708	306	1.047	<.001	0.904	0.068	0.076				
Step1 Gc/Gf	95.146	102	1.037	.671	1.000	0.000	0.041				
Step1 Gc/NFC	1071.525	479	1.022	<.001	0.913	0.060	0.074				
Step1 Gf/Gf	1223.396	619	1.019	<.001	0.908	0.054	0.078				
Step 2	1512.975	848	1.009	<.001	0.920	0.048	0.075				
Step 3	Not provided										
Step1 Gc/Gf/class	111.505	114	1.042	.549	1.000	0.000	0.041				
Step1 Gc/NFC/class	1138.506	509	1.021	<.001	0.910	0.060	0.074				
Step1 Gf/Gf/class	1282.722	653	1.018	<.001	0.906	0.053	0.078				
Step 2/class	1574.187	886	1.009	<.001	0.919	0.048	0.075				
Step 3/class	Not provided										

Note. Following [18]’s ([18]) recommendation, we used comparative fit indices (ΔCFI) to compare two models of different measurement invariance levels. ΔCFI values of 0.01 or less were interpreted as a tolerable deterioration in model fit. NFC partial scalar = partial scalar measurement invariance across measurement points was achieved by allowing the intercept of item 5 to vary across measurement points.

**Table 4 jintelligence-12-00104-t004:** Results of Hypotheses Tests in the Present Study.

Model	H1 Gc → ΔGc	H2 Gf → ΔGc	H3 NFC → ΔGc	RQ4 Gf × NFC → ΔGc	RQ5 Gc → ΔGf	RQ6 Gf → ΔGf	H7 NFC → ΔGf	RQ8 Gc → ΔNFC	H9 Gf → ΔNFC	RQ10 NFC → ΔNFC
Step 1	−	0	0		0	0	+	+	+	−
Step 2	−	0	0		0	0	+	0	+	−
Step 3	−	0	0	0	0	0	+	0	+	−

Note. + indicates that a significant positive prediction path was observed. 0 indicates that the prediction was investigated but no significant path was found. − indicates that a significant negative prediction path was observed. In Step 1, the LCSs of any two constructs were observed simultaneously (i.e., NFC and Gf; NFC and Gc; Gf and Gc). In Step 2, a single model was conducted containing all three variables. In Step 3, this model was extended to include the latent interaction score of Gf and NFC at T1.

## Data Availability

Analysis code is available online https://doi.org/10.17605/OSF.IO/9547T (Accessed on 20 October 2024). Due to legal rights, data cannot be made publicly available. Data will be provided upon request.

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
