# Peer review of "Crystallized Intelligence, Fluid Intelligence, and Need for Cognition: Their Longitudinal Relations in Adolescence"

_jintelligence, 2024, doi:10.3390/jintelligence12110104_

Round 1

Reviewer 1 Report

Comments and Suggestions for Authors

In this manuscript, the authors investigate the reciprocal relations between fluid and crystallized intelligence and need for cognition in a sample of 341 German adolescent students. The methods employed in the study are sound, and the introduction provides both an interesting and comprehensive review of existing studies in the field, as well as a nice theoretical embedding as a basis for hypotheses and analyses.

My primary concern, however, is the novelty of the findings. The authors need to work on how they highlight what’s unique to the present study. It is no compelling argument to point out the inconclusiveness and heterogeneity of previous research (as on p.9) when the current study ‘only’ adds another sample and another set of results. I do not question whether this research can be a valuable contribution (it surely can), but the framing needs to be improved. In my opinion, the main selling points are the investigation of all reciprocal paths between the constructs and the inclusion of the latent interaction between Gf and NFC (even though the latter was no significant predictor of Gc).

In the following, I will outline  further issues which the authors might want to address before the manuscript can be considered fully eligible for publication in Journal of Intelligence.

1.       I think it’s no good choice to mention the negative prediction of change in Gc by its initial level as the first bullet point as one plausible interpretation is regression to the mean, which is included in the discussion.

2.       You have relatively many hypotheses and therefore, it may be hard for some readers to follow. It would be helpful to add explanations on which theory the hypotheses derived from directly to the hypotheses.

3.       Also, please label research questions RQ instead of H and list them after the hypotheses. This makes even more sense since they were added after preregistration.

4.     46% of the sample stems from special classes for the gifted and the rest from regular classes. Does this have any relevance for the present findings? I can see that it is probably not possible to split the sample due to the sample size, but could you somehow control for it? At least you should mention it in the discussion.

5.       When reading your preregistration, it is noticeable that you planned on including academic interest in the analyses. Please explain why interest is not in the final manuscript, when you intended to investigate it.

6.       Did you include any control variables? Please justify your choice in the manuscript.

7.       I couldn’t find any model fit indices. Please add them if the models were not just identified.

8.       In addition, latent interaction models require a rather large sample size and can get shaky from time to time. (This might be one reason for the Heywood cases you found in the Step 3 model.) You might want to add a comment on this aspect in the discussion (e.g., on p. 19 where you already discuss the sample size).

9.       You did not provide any information on the variance of the change factors. In other words, is there anything to predict? Please add this.

10.   H8: The found effect disappeared when controlling for Gf. Nevertheless, you elaborate rather extensively on this effect. It might be more sensible to focus more on the potential role of Gf in this part of the discussion (p.18)

11.   Figure 3. The headline is wrong; it should say “Step 3”.

Author Response

Reviewer 1

Comments and Suggestions for Authors

Comments: In this manuscript, the authors investigate the reciprocal relations between fluid and crystallized intelligence and need for cognition in a sample of 341 German adolescent students. The methods employed in the study are sound, and the introduction provides both an interesting and comprehensive review of existing studies in the field, as well as a nice theoretical embedding as a basis for hypotheses and analyses.

Response: Thank you for this overall positive evaluation.

Comments: My primary concern, however, is the novelty of the findings. The authors need to work on how they highlight what’s unique to the present study. It is no compelling argument to point out the inconclusiveness and heterogeneity of previous research (as on p.9) when the current study ‘only’ adds another sample and another set of results. I do not question whether this research can be a valuable contribution (it surely can), but the framing needs to be improved. In my opinion, the main selling points are the investigation of all reciprocal paths between the constructs and the inclusion of the latent interaction between Gf and NFC (even though the latter was no significant predictor of Gc).

Response: Thank you for this valuable suggestion, which helped us to improve the "Present Study" section. We now emphasize that the provided comprehensive overview of previous longitudinal research on the interplay between Gc, Gf, and NFC highlights the inconsistency and heterogeneity of findings and the need for a study that examines all possible pathways simultaneously. We emphasize that our study examines all reciprocal pathways between Gc, Gf, and NFC over time, providing a complete picture of relations. In addition, we highlight the inclusion of the interaction between Gf and NFC, a relationship that has only been examined in one study to date (Lechner, 2019), demonstrating the novel aspect of our approach.

In the following, I will outline  further issues which the authors might want to address before the manuscript can be considered fully eligible for publication in Journal of Intelligence.

Comments 1: I think it’s no good choice to mention the negative prediction of change in Gc by its initial level as the first bullet point as one plausible interpretation is regression to the mean, which is included in the discussion.

Response 1: Thank you for this comment. We agree that this should not be a bullet point, as different interpretations of this effect are possible. In the updated version, the bullet point now simply states that changes in Gc were not predicted by the initial levels of fluid intelligence or need for cognition.

Comments 2: You have relatively many hypotheses and therefore, it may be hard for some readers to follow. It would be helpful to add explanations on which theory the hypotheses derived from directly to the hypotheses.

Response 2: Thank you. In the revised manuscript, we have added a brief explanation for each directional hypothesis, referencing the theoretical basis from which it was derived.

Comments 3: Also, please label research questions RQ instead of H and list them after the hypotheses. This makes even more sense since they were added after preregistration.

Response 3: Thank you for this suggestion. In the revised manuscript, we now use research questions (RQs) and indicate when a RQ includes a hypothesis, providing the theoretical rationale in each case. For RQ without directional hypothesis, we have explicitly labeled them as open research questions. In addition, we indicated for each RQ whether it was pre-registered or added post hoc.

We have maintained the original order of the 10 RQ to preserve the logical structure of the analyses in our study. Note that the RQ are organized by first examining all pathways related to changes in Gc, followed by those related to changes in Gf, and finally those related to changes in NFC. We find this order both conceptually coherent and consistent with the analytic approach we used and therefore decided to keep it.

Comments 4:     46% of the sample stems from special classes for the gifted and the rest from regular classes. Does this have any relevance for the present findings? I can see that it is probably not possible to split the sample due to the sample size, but could you somehow control for it? At least you should mention it in the discussion.

Response 4: Thank you for this suggestion. Indeed, we had not initially considered possible effects of class type, as the differences between regular and gifted classes are beyond the scope of the present manuscript. Our sample size is not large enough for a multi-group comparison to estimate prediction paths separately for each group. However, we were able to follow your suggestion and carried out robustness analyses including class type as an additional predictor. These analyses are now reported in the Supplementary Materials. The results remained largely unchanged when class type was included in the model. That is, the significant paths remained significant and the insignificant paths non-significant. We have also addressed this point in the “Limitations” section of the manuscript.

Comments 5: When reading your preregistration, it is noticeable that you planned on including academic interest in the analyses. Please explain why interest is not in the final manuscript, when you intended to investigate it.

Response 5: Thank you for pointing this out. We preregistered to examine two investment traits, NFC and academic interest, and their longitudinal interplay with Gc and Gf. We decided to focus solely on NFC for the present study because it is a domain-general representation of investment traits. According to Von Stumm (2016), NFC can be interpreted as the general factor of different investment traits, making it an optimal candidate for examining the interplay between intelligence (i.e., Gf and Gc assessed as broad constructs in our study) and investment traits. In contrast, academic interests were assessed as domain-specific constructs. Based on Brunswik symmetry (Brunswik, 1952; see also Kretschmar et al., 2018), we decided not to include them in this study. However, we plan to explore interest in a separate study with domain-specific outcomes (academic achievement in the same domains for which interest was assessed). We have included this rationale in the manuscript for clarity.

Comments 6: Did you include any control variables? Please justify your choice in the manuscript.

Response 6: In this revision, we conducted robustness analyses including class type as an additional predictor in the model (see our response to Point 4). Apart from this, no other control variables were included. Including control variables can lead to variance restrictions, and our study's scope is not focused on detecting effects under controlled conditions with the aim to find causal explanations, but rather on describing the longitudinal relationships between Gc, Gf, and NFC. We have clarified this point in the new Robustness Analyses section and added it to the limitations section.

Comments 7: I couldn’t find any model fit indices. Please add them if the models were not just identified.

Response 7: Thank you for pointing this out. We have added model fit indices for all models in Table 3.

Comments 8: In addition, latent interaction models require a rather large sample size and can get shaky from time to time. (This might be one reason for the Heywood cases you found in the Step 3 model.) You might want to add a comment on this aspect in the discussion (e.g., on p. 19 where you already discuss the sample size).

Response 8: Thank you for the suggestion. We agree that the limited sample size could be a plausible explanation for the Heywood cases. We have added this point to the Limitations section.

Comments 9: You did not provide any information on the variance of the change factors. In other words, is there anything to predict? Please add this.

Response 9: Thank you for pointing this out. We included this information in the “Latent Change Score Analyses” section. There was significant variation in the LCS for all variables (Gc: Variance = 1.15, p = .001; Gf: Variance = 8.37, p = .035; NFC: Variance = 0.23, p < .001).

Comments 10: H8: The found effect disappeared when controlling for Gf. Nevertheless, you elaborate rather extensively on this effect. It might be more sensible to focus more on the potential role of Gf in this part of the discussion (p.18)

Response 10: Thank you for this suggestion. We agree with your evaluation and have revised the discussion of RQ8 to place more emphasis on the potential role of Gf. We have also acknowledged the need for further research to explore the potential mediating role of Gf in this relationship.

Comments 11: Figure 3. The headline is wrong; it should say “Step 3”.

Response 11: Thank you for pointing this out; we changed the headline accordingly.

Reviewer 2 Report

Comments and Suggestions for Authors

The submitted manuscript reports on a study that examined the relationships between crystallized intelligence, fluid intelligence, and need for cognition measured at two time points among adolescents. In sum, they examined whether each variable predicted change in each of the other variables and in itself. This manuscript has several strengths. It is well-written, and the authors thoroughly reviewed the literature and presented several relevant theories.

Although it is a plus that this study has more than one measurement time, it is still not optimal that there are only 2 measurement occasions (see Ployhart & MacKenzie, 2015). The authors note that having at least 3 measurement occasions would allow for other tests (e.g., mediation). But more importantly is that having at least three measurement occasions would address the possible regression to the mean. As the authors note, that initial crystallized intelligence scores negatively predicted change in crystallized intelligence could be that high scorers at time 1 regressed to the mean at time 2. Given this, I think that the authors should discuss how only having 2 measurement occasions was one of the study limitations.

It would be informative to provide more detail about recruitment and the description of the study provided to the participants. That the participants spent 5 hours at each wave is extraordinary. What was the incentive for the students to participate? That time length suggests to me that the intelligence tests were part of school curriculum and that the researchers added the self-report measures.

It would also be helpful to provide additional information about the procedures. How many participants were in each testing group? Who administered the tests? Were they the teachers or researchers? If they were researchers, were they undergraduate students, graduate students, or the principal investigators? In addition, how many students were in each testing group?

It would be informative to compare the participants from regular classes to those from gifted class on key variables. Moreover, it would be informative to address variability due to the school attended.  

In the Abstract, the authors only mentioned two statistically significant effects. They did not note that most of their hypotheses were not supported. As such, it was unexpected to get to the results and learn that the findings were more complicated than suggested in the Abstract.

Reference

Ployhart, R. E., & MacKenzie, W. I., Jr. (2015). Two waves of measurement do not a longitudinal study make. In C. E. Lance & R. J. Vandenberg (Eds.), More statistical and methodological myths and urban legends. (pp. 85–99). Routledge/Taylor & Francis Group.

Author Response

Reviewer 2

Comments and Suggestions for Authors

Comments 1: The submitted manuscript reports on a study that examined the relationships between crystallized intelligence, fluid intelligence, and need for cognition measured at two time points among adolescents. In sum, they examined whether each variable predicted change in each of the other variables and in itself. This manuscript has several strengths. It is well-written, and the authors thoroughly reviewed the literature and presented several relevant theories.

Response 1: Thank you very much for the overall positive evaluation of our work!

Comments 2: Although it is a plus that this study has more than one measurement time, it is still not optimal that there are only 2 measurement occasions (see Ployhart & MacKenzie, 2015). The authors note that having at least 3 measurement occasions would allow for other tests (e.g., mediation). But more importantly is that having at least three measurement occasions would address the possible regression to the mean. As the authors note, that initial crystallized intelligence scores negatively predicted change in crystallized intelligence could be that high scorers at time 1 regressed to the mean at time 2. Given this, I think that the authors should discuss how only having 2 measurement occasions was one of the study limitations.

Response 2: Thank you for pointing this out. We have now included this point to the Limitations section and addressed the fact that with only two measurement occasions it is more difficult to distinguish between true change and regression to the mean. We also added that three or more measurement occasions would allow for more accurate modeling of growth parameters (Ployhart & MacKenzie, 2015) and the investigation of nonlinear developmental effects. We outlined this as a research question for future research.

Comments 3: It would be informative to provide more detail about recruitment and the description of the study provided to the participants. That the participants spent 5 hours at each wave is extraordinary. What was the incentive for the students to participate? That time length suggests to me that the intelligence tests were part of school curriculum and that the researchers added the self-report measures.

Response 3: Thank you for raising this point. We were fortunate to be able to conduct these tests exclusively for research purposes, rather than as part of the school curriculum. As an incentive, students received personalized feedback on their ipsative intelligence and interest profiles in a letter at the end of the study. In addition, schools received a report summarizing the aggregated results of their students. We have added this information to the Participants and Procedures section.

Comments 4: It would also be helpful to provide additional information about the procedures. How many participants were in each testing group? Who administered the tests? Were they the teachers or researchers? If they were researchers, were they undergraduate students, graduate students, or the principal investigators? In addition, how many students were in each testing group?

Response 4: Thank you for pointing out that this information was missing. The group size during testing ranged from 2 to 26 students (M = 12.73, SD = 6.61). The tests were administered by trained student assistants (undergraduates) and principal researchers, with two administrators present per group. We have added this information to the Participants and Procedures section.

Comments 5: It would be informative to compare the participants from regular classes to those from gifted class on key variables. Moreover, it would be informative to address variability due to the school attended. 

Response 5: Thank you for this suggestion. Reviewer 1 raised a similar point related to class type. While we had not initially considered the effects of class type, as the differences between regular and gifted classes are beyond the scope of this manuscript, we acknowledge their potential impact. Unfortunately, our sample size is not large enough for a multi-group comparison to estimate prediction paths separately for each class type. However, we conducted robustness analyses including class type as an additional predictor to test whether this affected the results. These results indicate that the significant paths remained significant, and the non-significant paths remained unchanged (see Supplementary Materials). In addition, we now report manifest correlations and descriptive statistics separately for regular and gifted classes in Table S1 of the Supplementary Materials.

Variability due to the school attended and the specific class attended creates a nested data structure that can potentially bias the results. The students attended four different schools with 36 different classes. Dividing the dataset by school would result in sub-datasets that are too small for separate analyses, and having only four schools is insufficient to statistically account for this nested structure. However, it was possible to describe and control for the influence of class differences. We have now added ICC values for each variable reflecting class differences in Table 2. The ICCs for all variables were substantial (ICC > .10), especially for Gc (T1: ICC = .44; T2: ICC = .47). This may be because we included both regular and gifted classes into our analyses.

We accounted for these differences in two ways. First, all SEM analyses were conducted using the 'type = complex' command, which adjusts standard errors for class-level differences. Second, we conducted a robustness analysis that included class type as a control variable (see above).

Comments 6: In the Abstract, the authors only mentioned two statistically significant effects. They did not note that most of their hypotheses were not supported. As such, it was unexpected to get to the results and learn that the findings were more complicated than suggested in the Abstract.

Response 6: Thank you for this comment. We have revised the abstract to more accurately reflect our findings, including the fact that, contrary to our theoretical assumptions, changes in Gc were not significantly predicted by initial Gf, initial NFC, or their interaction.

Reference

Ployhart, R. E., & MacKenzie, W. I., Jr. (2015). Two waves of measurement do not a longitudinal study make. In C. E. Lance & R. J. Vandenberg (Eds.), More statistical and methodological myths and urban legends. (pp. 85–99). Routledge/Taylor & Francis Group.

Round 2

Reviewer 1 Report

Comments and Suggestions for Authors

Thank you for responding so thoroughly to my responses. The manuscript has been substantially improved and is now ready for publication.

Just one very minor point, relating to my comment 3, in which I probably wasn't clear enough: I didn't mean you should rename all hypotheses into research questions, only those without a clear expectation (RQ 4, 5, 8, and 10). The others are hypotheses and can be named as such. Thank you.

Author Response

Reviewer comments: Thank you for responding so thoroughly to my responses. The manuscript has been substantially improved and is now ready for publication.

Just one very minor point, relating to my comment 3, in which I probably wasn't clear enough: I didn't mean you should rename all hypotheses into research questions, only those without a clear expectation (RQ 4, 5, 8, and 10). The others are hypotheses and can be named as such. Thank you.

Reply: Thank you for pointing this out. We have changed this accordingly in all relevant sections of the manuscript.

Reviewer 2 Report

Comments and Suggestions for Authors

The authors addressed my concerns. This work makes a contribution to the literature.

Author Response

Reviewer comments: The authors addressed my concerns. This work makes a contribution to the literature.

Reply: Thank you very much for this positive evaluation of our work.